# EFFICIENT MIP-LP GAP MITIGATION FOR PREDICT+OPTIMIZE

## ABSTRACT

The Predict+Optimize (P+O) paradigm seeks to train prediction models for unknown parameters in optimization problems, with the goal of yielding good optimization solutions downstream. Prior works have proposed strategies for gradient computation in neural network training, when the downstream optimization is a linear program (LP). Yet, in face of *mixed-integer* linear programs (MIP), much prior work simply relax the MIP into an LP, resulting in sub-optimally trained predictors. The issue is particularly stark in the recent Two-Stage Predict+Optimize framework, where even the MIP *constraints* can contain uncertainty.

In this work, we propose a (shockingly) simple and fast approach for addressing the MIP-LP gap, and show that it yields essentially the same or more accuracy gains over a much slower method adapted from prior work. Concretely, for the latter, we adapt the approach of MIPaaL (Ferber et al., 2020) and introduce cutting planes into the LP relaxation, before using LP-based gradient computation methods. Such adaptation is slow and requires some work for the new Two-Stage P+O setting, given the constantly-changing constraint predictions during training. We instead propose and advocate for a far simpler method: replace the relaxed-LP optimum in the LP-based gradient computation with the actual true MIP optimum, avoiding the repeated use of (slow) cutting plane MIP solvers in the slow method.

Experimental results on 3 benchmarks show that this simple strategy yields the same or more accuracy gain over the much slower cutting plane approach, and the conjunctive use of the two methods yields only minor further gains at the expense of vastly increased training time, sometimes by a whole order of magnitude.

## 1 INTRODUCTION

Predict+Optimize (Elmachtoub & Grigas, 2022) is a recently proposed paradigm at the intersection of constrained optimization and machine learning. In a constrained optimization problem, suppose some parameters are unknown, with some relevant features that can help us *predict* these unknown parameters. How do we make best use of the training data to train a prediction model (e.g. a neural network) that gives good parameter predictions, so as to yield an estimated solution that has good objective even under the true unknown parameters? The key idea by Elmachtoub & Grigas (2022) is to incorporate the optimization problem into the definition of the training (and test) loss, producing an optimization-aware prediction model.

Since the advent of this Predict+Optimize paradigm, a number of works have sought to derive gradient computation methods for training neural network prediction models (Mandi & Guns, 2020; Amos & Kolter, 2017; Paulus et al., 2021). Yet, most of these works assume linear programs, quadratic programs or other convex programs for the underlying downstream optimization tasks. For (mixed) integer programs and other discrete non-convex programs commonly used in AI and operations research applications, many (but not all) prior works simply ask the practitioner to simply discard the integrality constraints and use continuous-optimization based training methods. Clearly this is sub-optimal, as these training methods and the resulting neural network models completely ignore the integrality aspects of the optimization problem. The issue became more pressing in the recent Two-Stage Predict+Optimize framework (Hu et al., 2023; 2022a), which allows uncertainty and unknown parameters to appear in the optimization *constraints*, instead of only in the optimization objective as in the original Predict+Optimize framework.

In this work, we study two strategies that incorporate the integrality constraints back into training.

1. One strategy is to adapt an idea from MIPaaL (Ferber et al., 2020), which is to add cutting planes to the LP-relaxed polytope to make it closer to the integral optimization domain, before computing gradient information from this tightened relaxation. Doing so requires using a cutting plane MIP solver to generate cutting planes, which is slower than some other types of MIP solvers. While MIPaaL only handled predicting unknown objective parameters, in our context we have to accommodate the entire Two-Stage Predict+Optimize framework with unknown constraint parameters as well. As training proceeds, the constraint predictions change, necessitating new cuts to be generated as the old ones might be invalid. This strategy thus adds dominating training time overhead, and Section 3.1 explores ways to limit cut generation to trade accuracy gains for running time.

2. **The novel strategy we propose and advocate is much simpler and faster**, by avoiding the use of cutting plane MIP solvers. Starting with known gradient computation methods for LP (relaxation) models, we replace the LP optimum with the true MIP optimum in the calculations and use the resulting gradient. While this simple strategy still requires solving MIPs, we no longer need to use slow cutting plane solvers and can instead use much faster solvers based on branch-and-bound search. Section 3.2 discusses the method details.

Across 3 experimental benchmarks, we demonstrate that the very simple second strategy yields prediction accuracy that is essentially at least as good as the slow cutting plane strategy, while being much faster, in some cases by an order of magnitude. The further accuracy gains from using both strategies are also minimal. Together, our results strongly suggest that our novel and simple approach of just using the MIP optimum is a practical solution to the Predict+Optimize problem on MIPs.

**Other related work**  Elmachtoub and Grigas first proposed the Predict+Optimize framework (Elmachtoub & Grigas, 2022), with numerous followup work in the community on improving computational efficiency (Mandi et al., 2020; Mulamba et al., 2021), predictive accuracy (Demirović et al., 2020; Jeong et al., 2022; Mandi & Guns, 2020), and on types of applicable optimization problems (Guler et al., 2022; Hu et al., 2022b; Wilder et al., 2019). Other works also apply the framework to specific real-world scenarios (Chu et al., 2023; Stratigakos et al., 2022; Tian et al., 2023). More recently, Hu et al. (2022a; 2023; 2024) proposed adaptations of the framework to handle uncertainty in optimization constraints, including the Two-Stage framework which our work is most related to.

See Appendix E for further related work, e.g. under the umbrella of *decision-focused learning*.

## 2 BACKGROUND

In this Background section, we recount the Two-Stage Predict+Optimize framework (Hu et al., 2023), as well as explain their gradient computation method adapted from IntOpt (Mandi & Guns, 2020).

### 2.1 TWO-STAGE PREDICT+OPTIMIZE

Without loss of generality, the downstream optimization is a minimization problem. The exposition below is mostly from Hu et al. (2023); see their Sections 2 and 3 for a detailed explanation/examples.

A *parameterized optimization problem (Para-OP)* $P(\theta)$ is defined as computing:

$$x^*(\theta) = \arg \min_x obj(x, \theta) \quad \text{s.t. } C(x, \theta)$$

where $x$ is a vector of decision variables, $\theta$ is a vector of parameters, $obj$ is a function mapping decisions $x$ and parameters $\theta$ to a real objective value that is to be minimized, and $C$ is a set of constraints that must be satisfied over $x$ under parameters $\theta$. We call $x^*(\theta)$ an *optimal solution* under parameters $\theta$, and $obj(x^*(\theta), \theta))$ the *optimal value* under parameters $\theta$. When the parameters are all known, a Para-OP is just a classical optimization problem (OP).

Each instantiation of the true parameter vector $\theta$ has an associated *feature matrix* $F$. These features are relevant and correlated with the unknown parameters $\theta$, which can help a model predict $\theta$.

After defining Para-OPs, we can now fully describe the Two-Stage Predict+Optimize framework.

**Stage 1** The practitioner uses a prediction model, which takes in a feature matrix $F$, to compute a vector of estimated parameters $\hat{\theta}$. The Stage 1 solution $\hat{x}^{(1)}$ is then computed as

$$\hat{x}^{(1)} = \arg\min_{x} obj(x, \hat{\theta}) \quad \text{s.t. } C(x, \hat{\theta})$$

The solution $\hat{x}^{(1)}$ should be interpreted as a *soft commitment* modifiable in Stage 2, under penalty.

**Stage 2** The true parameters $\theta$ are revealed, and the practitioner wishes to compute an updated Stage 2 solution $\hat{x}^{(2)}$ from $\hat{x}^{(1)}$, subject to a (problem+application-specific) penalty function $Pen(\hat{x}^{(1)} \to \hat{x}^{(2)}, \theta)$ which depends on both the softly-committed Stage 1 $\hat{x}^{(1)}$, the final Stage 2 solution $\hat{x}^{(2)}$ and the true parameters $\theta$. More specifically, the Stage 2 solution $\hat{x}^{(2)}$ is computed as

$$\hat{x}^{(2)} = \arg\min_{x} obj(x, \theta) + Pen(\hat{x}^{(1)} \to x, \theta) \quad \text{s.t. } C(x, \theta)$$

The Stage 2 solution $\hat{x}^{(2)}$ should be interpreted as a hard-committed final action, and note that it is guaranteed to be feasible under the true parameters $\theta$.

The prediction $\hat{\theta}$ is evaluated using the *post-hoc regret*, which is the sum of two terms: (a) the difference in objective between the *true optimal solution* $x^*(\theta)$ and the final Stage 2 solution $\hat{x}^{(2)}$ under the true parameters $\theta$, and (b) the penalty incurred by modifying $\hat{x}^{(1)}$ to $\hat{x}^{(2)}$. Formally, the post-hoc regret function $PReg(\hat{\theta}, \theta)$ (for minimization problems) is:

$$PReg(\hat{\theta}, \theta) = obj(\hat{x}^{(2)}, \theta) - obj(x^*(\theta), \theta) + Pen(\hat{x}^{(1)} \to \hat{x}^{(2)}, \theta)$$

The goal of a prediction model is to make predictions $\hat{\theta}$ so as to minimize the post-hoc regret. The post-hoc regret, crucially defined in terms of the underlying optimization, is thus used as the training loss for the prediction model, in order to produce an optimization-aware prediction at test time.

## 2.2 Gradient computation for LP models

Much prior work in Predict+Optimize have focused on training neural networks as the prediction model, which is also the goal in the present work. The standard training techniques are by gradient methods and backpropagation, which compute the gradient of the post-hoc regret with respect to changes in the network edge weights. By the law of total derivatives, the gradient decomposes to

$$\frac{dPReg(\hat{\theta}, \theta)}{dw_e} = \frac{\partial PReg(\hat{\theta}, \theta)}{\partial \hat{x}^{(2)}}\bigg|_{\hat{x}^{(1)}} \frac{\partial \hat{x}^{(2)}}{\partial \hat{x}^{(1)}} \frac{\partial \hat{x}^{(1)}}{\partial \hat{\theta}} \frac{\partial \hat{\theta}}{\partial w_e} + \frac{\partial PReg(\hat{\theta}, \theta)}{\partial \hat{x}^{(1)}}\bigg|_{\hat{x}^{(2)}} \frac{\partial \hat{x}^{(1)}}{\partial \hat{\theta}} \frac{\partial \hat{\theta}}{\partial w_e} \quad (1)$$

The key challenge lies in the $\frac{\partial \hat{x}^{(1)}}{\partial \hat{\theta}}$ and $\frac{\partial \hat{x}^{(2)}}{\partial \hat{x}^{(1)}}$ terms, which are gradients of the optima of the Stage 1 and 2 optimizations with respect to their parameters (e.g. $\hat{\theta}$ and $\hat{x}^{(1)}$). Even if the optimization problems are as continuous and convex as LPs, the gradients are 0 almost everywhere, leading to no useful training gradients. The problem is further exacerbated for MIPs, with integrality constraints.

Here, we present the method proposed by Hu et al. (2023), which extends IntOpt (Mandi & Guns, 2020) to handle unknown parameters in constraints. The method is designed to "differentiate through" an LP, if both Stage 1 and 2 optimizations can be expressed as LPs—for MIPs, both prior works just take the LP relaxation. The goal of this current paper is to propose a novel simple strategy (Section 3.2) to *efficiently* re-incorporate the integrality constraints into the training process.

Without loss of generality, consider a minimization LP of the following standard form:

$$x^* = \arg\min_{x} c^\top x \quad \text{s.t. } Ax = b, Gx \geq h, x \geq 0 \quad (2)$$

As explained above, if we try to take the gradient of $x^*$ with respect to any of the parameters $c$, $A$, $b$, $G$ or $h$, the gradients will be 0 almost everywhere (over the tuples of $(c, A, b, G, h)$).

Following the interior-point based approach of IntOpt (Mandi & Guns, 2020) and (Hu et al., 2023), consider the following relaxation using logarithmic regularizer terms, for any fixed value $\mu \geq 0$:

$$x^* = \arg\min_{x,s} c^\top x - \mu \sum_{i=1}^{d} \ln(x_i) - \mu \sum_{i=1}^{q} \ln(s_i)$$
$$\text{s.t. } Ax = b, Gx - s = h \quad (3)$$

This relaxes (2) by introducing slack variables $s \geq 0$ to re-express $Gx \geq h$ into $Gx - s = h$, and moves the non-negativity constraints $x \geq 0$ and $s \geq 0$ into the logarithmic barrier terms in the objective, with multiplier $\mu \geq 0$. We choose the hyperparameter $\mu$ as $10^{-3}$, following Hu *et al.*

Given this relaxation (3), Slater's condition holds, and so the KKT conditions must be satisfied at the primal optimum $(x^*_{\text{int}}, s^*_{\text{int}})$ with some dual optimum $y^*_{\text{int}}$.

The remaining part, then, is to compute all the relevant gradients using the KKT conditions. The implicit function theorem lets us differentiate the KKT condition equations and express the gradients as solutions to a system of linear equations. By using an interior point solver (Mandi & Guns, 2020) on the relaxed problem (3), we can find the primal-dual optimum solutions $((x^*_{\text{int}}, s^*_{\text{int}}), y^*_{\text{int}})$, and then solve for the desired gradients by substituting the optimal solution values into the KKT-derivative linear system.

The precise calculation details are not essential for understanding the contributions of this paper, so we omit these, and refer interested readers to Hu et al. (2023).

## 3 TRAINING ALGORITHMS BEYOND VANILLA LP RELAXATIONS

In Section 2.2, we saw the prior approach for differentiating through MIPs in the Two-Stage Predict+Optimize setting, which was just ignoring integrality constraints in the MIP and using an approach for differentiating through the corresponding LP relaxation. This is clearly sub-optimal — the gradient computation, and hence the entire neural network training process, completely ignores the fact that the downstream optimization is a MIP and not an LP. To mitigate this deficiency, in this section we give an adaptation from prior work (Section 3.1) and further propose a novel, much simpler and much faster strategy (Section 3.2). We also choose the best configurations for both strategies, using small-scale experiments. Afterwards, in Section 4, we experimentally compare them to show that our novel and simple strategy (vastly) outperforms the adaptation from prior work, with no worse prediction accuracy (and sometimes much better), and always significantly faster training.

### 3.1 ADDING CUTTING PLANES TO LP RELAXATIONS

The first basic (semi-baseline) approach follows an idea from MIPaaL (Ferber et al., 2020), which is to add cutting planes to tighten the LP relaxation, to make it approximate the true MIP well (or perfectly). While MIPaaL was originally proposed for Predict+Optimize (and related) settings where unknown parameters appear only in the objective, in this work, we apply this idea to the newer Two-Stage Predict+Optimize setting which allows also for unknown constraint parameters.

**Cutting plane solvers** A cutting plane MIP solver iteratively solves LP relaxations; if a non-integral optimal solution is found, then the solver generates a cut (an inequality $a \cdot x \leq b$) that removes the found solution (which cannot be a MIP solution given non-integrality) but which is also guaranteed to not remove *any* MIP feasible solution (i.e. the generated cut is *valid* for the MIP). The cut is added to the constraint set, and the process is repeated until an integer optimum is found.

Thus, in gradient computations, one can use the final LP with all the cuts generated by the cutting plane solver, to achieve a much better approximation of the original MIP.

The original work of MIPaaL considered only Para-OPs with unknown objective parameters, without any uncertainty in the constraints. Thus, every generated cut remains a valid cut for the feasible region of the underlying MIP, regardless of how the objective parameter predictions change. On the other hand, when there is uncertainty in the constraints, directly using this cutting plane idea introduces a major complication—during training, the constraint parameter predictions are constantly changing, meaning that the feasible solution set of the MIP is also changing. This means that cuts generated for one constraint prediction may no longer be valid once the constraint prediction is updated (i.e. when we take a gradient step in the neural network training). As a result, cuts need to be generated from scratch after taking every gradient step, requiring numerous calls to a cutting plane MIP solver during the model training process. Given that cutting plane MIP solvers are generally much slower than other types of solvers, for example those based on branch-and-bound strategies, the repeated cutting plane solver calls add significant (and in some cases drastic) overhead to the training time.

Table 1: Question 3.1.1: mean **post-hoc-regret** and standard deviations (not confidence intervals) over 10 simulations on the weighted set multi-cover problem.

| Penalty factor | LP | FullCut-Stage1 | FullCut-Both |
|---|---|---|---|
| 0.25 | 50.68±18.29 | 40.65±16.15 | 38.81±10.87 |
| 0.5 | 71.20±20.47 | 67.91±18.33 | 65.44±19.01 |
| 1 | 113.84±31.42 | 109.93±31.80 | 104.57±31.59 |
| 2 | 175.63±51.87 | 167.87±45.94 | 161.89±46.47 |
| 4 | 275.09±70.54 | 244.07±68.43 | 241.33±68.43 |

Table 2: Question 3.1.1: mean **training time** (in seconds) and standard deviations (not confidence intervals) over 10 simulations on the weighted set multi-cover problem.

| Penalty factor | LP | FullCut-Stage1 | FullCut-Both |
|---|---|---|---|
| 0.25 | 91.93±22.98 | 1192.21±598.81 | 1936.23±630.23 |
| 0.5 | 89.05±33.21 | 1197.12±680.50 | 1795.77±912.37 |
| 1 | 100.15±54.98 | 1564.64±884.82 | 2777.01±1449.29 |
| 2 | 78.87±27.02 | 1401.44±767.93 | 2481.50±1033.75 |
| 4 | 61.74±14.17 | 954.75±411.10 | 2035.98±784.50 |

The rest of this subsection investigates techniques to reduce this overhead, perhaps at the expense of some prediction accuracy, but which are still better than not using integrality information at all.

Recalling Equation (1), there are two derivative terms requiring differentiating through a MIP: $\frac{\partial \hat{x}^{(1)}}{\partial \hat{\theta}}$ and $\frac{\partial \hat{x}^{(2)}}{\partial \hat{x}^{(1)}}$. These are derivatives corresponding to the Stages 1 and 2 optimizations respectively. This brings us to the first question on the application of the cutting plane strategy.

**Question 3.1.1**  Should we generate cuts for the optimizations in both Stages 1 and 2? Or is there a better tradeoff, for example, by only generating cuts for Stage 1?

Beyond the decision on whether to restrict cutting plane generation to Stage 1 gradient computation, even when differentiating through a MIP for a particular stage, there is the question of whether to fully generate cuts until the LP relaxation has the same optimum as the MIP.

**Question 3.1.2**  When differentiating through a MIP, is it worth running the cutting plane solver until completion? Can we just generate the first few cuts, or alternatively, limit the cut generation time, and use the resulting LP relaxation instead?

Here, we give some preliminary results, based on the "weighted set multi-cover" problem described in Section 4, addressing the above questions on the best use of the cutting plane strategy. Section 4 will then compare the best cutting plane strategy with the Section 3.2 strategy with full experiments.

**Answer to Question 3.1.1**  Based on Tables 1 and 2, it is reasonable to generate cuts solely for Stage 1 (referred to as "FullCut-Stage1") rather than for both Stages 1 and 2 (referred to as "FullCut-Both"). The differences in post-hoc regret between the two approaches are minimal, while the time savings from focusing on Stage 1 are significant. Specifically, across all penalty factors, FullCut-Both yields post-hoc regrets at most 5% smaller than FullCut-Stage1, while the training time for FullCut-Stage1 is only roughly 60-70% of that for FullCut-Both. The results suggest focusing on generating cuts for Stage 1 only, for a good tradeoff.

**Answer to Question 3.1.2**  Tables 3 and 4 show that generating only the first 10 cuts (referred to as "LimCut-Stage1 (CutNum=10)") or limiting the cut generation time to 0.5s (referred to as "LimCut-Stage1 (CutGenTime=0.5s)") could be reasonable compromises between prediction accuracy versus training time. From Tables 3 and 4, we observe that LimCut-Stage1 (CutNum=10) and LimCut-Stage1 (CutGenTime=0.5s) generally perform well, achieving 3.21%-10.81%, and 2.92%-10.33% smaller post-hoc regret than LP without any cuts respectively, across different penalty factors.

Moreover, these methods require significantly less training time compared to FullCut-Stage1 while still maintaining competitive performance. Specifically, the training times for LimCut-Stage1 (Cut-Num=10) are roughly only 60-80% of that required for FullCut-Stage1, with slightly less than 1% larger post-hoc regret than FullCut-Stage1 across the majority of penalty factors.

Table 3: Question 3.1.2: mean **post-hoc-regret** and standard deviations (not confidence intervals) over 10 simulations on the weighted set multi-cover problem.

| Penalty factor | LP | LimCut-Stage1 (CutNum=5) | LimCut-Stage1 (CutNum=10) | LimCut-Stage1 (CutGenTime=0.5s) | FullCut-Stage1 |
|---|---|---|---|---|---|
| 0.25 | 50.68±18.29 | 46.70±18.60 | 45.20±16.68 | 46.24±17.28 | 40.65±16.15 |
| 0.5 | 71.20±20.47 | 69.34±19.85 | 68.04±18.32 | 68.02±17.40 | 67.91±18.33 |
| 1 | 113.84±31.42 | 112.46±33.61 | 110.19±34.64 | 110.52±30.47 | 109.93±31.80 |
| 2 | 175.63±51.87 | 172.14±48.80 | 169.43±45.59 | 170.07±47.84 | 167.87±45.94 |
| 4 | 275.09±70.54 | 251.68±68.01 | 246.52±66.45 | 246.68±66.19 | 244.07±68.43 |

Table 4: Question 3.1.2: mean **training time** (in seconds) and standard deviations (not confidence intervals) over 10 simulations on the weighted set multi-cover problem.

| Penalty factor | LP | LimCut-Stage1 (CutNum=5) | LimCut-Stage1 (CutNum=10) | LimCut-Stage1 (CutGenTime=0.5s) | FullCut-Stage1 |
|---|---|---|---|---|---|
| 0.25 | 91.93±22.98 | 820.42±376.89 | 859.33±301.84 | 788.26±300.46 | 1192.21±598.81 |
| 0.5 | 89.05±33.21 | 796.69±459.58 | 895.71±381.90 | 803.54±334.65 | 1197.12±680.50 |
| 1 | 100.15±54.98 | 855.74±336.33 | 900.17±392.17 | 880.17±353.09 | 1564.64±884.82 |
| 2 | 78.87±27.02 | 800.90±329.20 | 830.03±305.23 | 773.56±321.68 | 1401.44±767.93 |
| 4 | 61.74±14.17 | 755.12±297.70 | 792.03±306.41 | 723.19±346.35 | 954.75±411.10 |

**Summary for cutting plane strategy**   Given the very significant overhead in using the cutting plane strategy, in Section 4 we will only use it for Stage 1, and further limit cut generation time to 0.5s per instance. This configuration will be compared against the Section 3.2 strategy.

## 3.2 FAST+SIMPLE: USING THE TRUE MIP OPTIMUM

The basic strategy from the previous section and adapted from the prior work of MIPaaL increases the training time very significantly even when we use time-limiting techniques. By contrast, the novel strategy we propose in this section is much simpler and drastically faster to run. As we will see in Section 4, this new strategy has (essentially) at least as good post-hoc regret. **The conceptual message of this paper is to advocate this simple method over the slower prior MIPaaL method.**

To understand the new strategy, first recall the gradient computation method for LPs, in Section 2.2. There, an interior point LP solver solves a relaxation (Equation (3)) of the LP, for the primal-dual optimum pair $((x_{\text{int}}^*, s_{\text{int}}^*), y_{\text{int}}^*)$. The required gradients can then be computed by differentiating through the KKT condition, which holds at the primal-dual optimum, and solve the resulting linear system. Lastly, for MIPs, ignore the integrality constraints and use the above strategy for LPs.

Our new strategy is very simple: instead of using $x_{\text{int}}^*$ which is the primal optimum of the relaxation (Equation (3)), we use the true MIP optimum $x_{\text{MIP}}^*$ in place of $x_{\text{int}}^*$ in the above calculations.

Our method bears some resemblance to the "straight-through" gradient estimator (Bengio et al., 2013), but as far as we can tell from explicit, rigorous calculations using the chain rule, it cannot mathematically be viewed as an instance of "straight-through".

Note that, unlike the cutting plane strategy in Section 3.1, there is no requirement on the type of MIP solver used for this new True MIP Optimum strategy. As such, we can use a fast solver, for example those with branch-and-bound algorithms, avoiding the use of slow cutting plane MIP solvers.

Given this simple (yet, as we will see, effective) strategy, we again have to decide whether to use it for the optimizations in *both* stages or restrict the strategy to Stage 1.

**Question 3.2**   Should we employ the True MIP Optimum strategy for both Stages 1 and 2 optimizations? Or is there a better tradeoff, for example, by only using it on Stage 1?

Table 5: Question 3.2: mean **post-hoc-regret** and standard deviations (not confidence intervals) over 10 simulations on the weighted set multi-cover problem.

| Penalty factor | LP | MipOpt-Stage1 | MipOpt-Both | FullCut-Both |
|---|---|---|---|---|
| 0.25 | 50.68±18.29 | 38.72±10.77 | 38.54±10.67 | 38.81±10.87 |
| 0.5 | 71.20±20.47 | 66.44±18.67 | 66.03±18.49 | 65.44±19.01 |
| 1 | 113.84±31.42 | 108.91±31.07 | 107.84±31.42 | 104.57±31.59 |
| 2 | 175.63±51.87 | 165.06±47.17 | 163.71±44.60 | 161.89±46.47 |
| 4 | 275.09±70.54 | 242.83±68.22 | 241.65±69.93 | 241.33±68.43 |

Table 6: Question 3.2: mean **training time** (in seconds) and standard deviations (not confidence intervals) over 10 simulations on the weighted set multi-cover problem.

| Penalty factor | LP | MipOpt-Stage1 | MipOpt-Both | FullCut-Both |
|---|---|---|---|---|
| 0.25 | 91.93±22.98 | 123.85±31.62 | 143.94±52.41 | 1936.23±630.23 |
| 0.5 | 89.05±33.21 | 135.14±53.11 | 159.05±63.21 | 1795.77±912.37 |
| 1 | 100.15±54.98 | 168.01±73.49 | 183.32±69.28 | 2777.01±1449.29 |
| 2 | 78.87±27.02 | 159.64±69.72 | 170.31±65.54 | 2481.50±1033.75 |
| 4 | 61.74±14.17 | 114.59±32.74 | 137.22±43.45 | 2035.98±784.50 |

**Answer**   Our preliminary results (Tables 5 and 6), again on the weighted set multi-cover problem described in Section 4, show that using the True MIP Optimum strategy in both optimization stages does offer some small post-hoc regret improvement over only using the strategy in Stage 1. The training times do have overhead over the vanilla approach from Section 2.2, but they are nowhere near as drastic as for the cutting plane approach in Section 3.1.

For the Section 4 comparisons, we will use the True MIP Optimum strategy on both Stages 1 and 2.

## 3.3 COMBINING BOTH STRATEGIES

A reasonable further question to ask is: what happens if we use both strategies at the same time? In Section 3.1, the preliminary experiments informed us to use the cutting plane strategy without running the cutting plane MIP solver to completion. Instead, we take only the first few cuts, resulting in an LP that might have optimum different from the original MIP. We can thus additionally use the "True MIP Optimum" strategy, by running a faster MIP solver (to completion) and use the true MIP optimum in the gradient calculations. This further adds to the training time, but a priori, it might be possible that the combination of the strategies will produce a much better prediction model.

The experiments in Section 4 will demonstrate that the tradeoff is not worth it—the training time is even longer than the cutting plane method, and yet the post-hoc regret reductions are insignificant.

## 4 EXPERIMENTAL EVALUATION

In this section, we compare (1) the cutting plane strategy LimCut-Stage1 (CutGenTime = 0.5s), which we will refer to as 'LCGT-Stage1', (2) our novel true MIP optimum strategy 'MipOpt-Both', and (3) the combination of these two strategies, which we will refer to as 'Combination'. We use three benchmarks: weighted set multi-cover, 0-1 knapsack, and the nurse rostering problem.

We compare the proposed strategies with the prior Two-Stage Predict+Optimize method which only differentiates through the LP relaxation (Hu et al., 2023), and we use "LP" to denote it. "LP" is our only decision-focused baseline: to our knowledge, only two other decision-focused prior works can handle unknown constraints: CombOptNet and Nandwani et al. (2022). Hu et al. (2023) already compared "LP" with CombOptNet and demonstrated the latter's worse predictive performance. On the other hand, Nandwani et al. (2022) has no runnable code, and so we are unable to compare with them. Finally, for completeness, our experiments also compare with classical non-Predict+Optimize regression methods (as even weaker baselines). See Appendix B for these complete comparisons.

Both the proposed strategies and LP have hyperparameters, which we tune via cross-validation. We include the hyperparameter types and values in Appendix D. All models are trained with Intel(R) Xeon(R) CPU E5-2630 v2 @ 2.60GHz processors. For the cutting plane solver, we use CPLEX (IBM, 2022) configured as a pure cutting plane solver, and for the non-cutting plane solver, we use Gurobi (Gurobi Optimization, LLC, 2023).

The source code and data are available at: MIPLP_Gap_Mitigation_for_Predict+Optimize.

**Weighted Set Multi-Cover Problem**   Our first benchmark is the weighted set multi-cover (WSMC) with unknown coverage requirements, which is a covering integer program. Let $I$ be a set of items and $J$ be a set of covers. The parameter $a_{ij}$ is 1 if the cover $j$ can cover item $i$, and 0 otherwise. Item $i \in I$ must be covered by at least $d_i$ many sets, and the cost of selecting cover $j \in J$ is $c_j$. The weighted set multi-cover problem (WSMC) aims to satisfy the coverage constraints while minimizing the total cost. The prediction challenge here is that the exact coverage requirements for

Table 7: Full comparisons: mean **post-hoc-regret** and standard deviations (not confidence intervals) over 30 simulations on the weighted set multi-cover problem.

| Penalty factor | LP | MipOpt-Both | LCGT-Stage1 | Combination |
|---|---|---|---|---|
| 0.25 | 36.19±17.96 | 30.70±16.12 | 31.65±17.63 | 29.22±15.99 |
| 0.5 | 52.91±30.57 | 49.78±27.48 | 51.26±26.80 | 49.42±27.47 |
| 1 | 82.72±44.65 | 76.16±46.55 | 79.81±44.20 | 74.00±46.69 |
| 2 | 123.67±67.16 | 118.18±62.40 | 120.03±60.40 | 117.31±55.09 |
| 4 | 204.44±108.45 | 188.53±92.73 | 194.85±91.59 | 177.74±97.36 |

Table 8: Full comparisons: mean **training time** (in seconds) and standard deviations (not confidence intervals) over 30 simulations on the weighted set multi-cover problem.

| Penalty factor | LP | MipOpt-Both | LCGT-Stage1 | Combination |
|---|---|---|---|---|
| 0.25 | 101.98±37.03 | 132.03±36.43 | 797.63±204.17 | 891.25±330.14 |
| 0.5 | 121.67±43.74 | 157.79±44.21 | 834.65±308.92 | 905.69±367.42 |
| 1 | 116.18±41.09 | 171.51±59.56 | 971.11±363.87 | 1006.51±398.06 |
| 2 | 79.76±24.86 | 165.65±49.35 | 902.68±327.34 | 963.59±325.11 |
| 4 | 64.84±10.52 | 135.42±32.73 | 729.61±320.57 | 736.73±319.55 |

each item are unknown. After the coverage requirements are revealed, if the selected covers cannot satisfy coverage requirements, extra covers can be added with cost $(1 + \rho)c_j$.

Please refer to Appendix A.1 for the precise MIP models for the Stages 1 and 2 optimizations.

We conduct experiments on 10 items and 50 covers. We generate the item-cover incidence matrices following the method of Grossman & Wool (1997). The cover costs are uniformly randomly drawn from $[1, 100]$. Coverage requirements $d_i$ are the unknown parameters and need prediction. Given the lack of datasets specific to this benchmark, we follow a standard Predict+Optimize experimental approach (Hu et al., 2023; Mulamba et al., 2021; Demirović et al., 2020) and use real data from a different problem (the ICON scheduling competition (Simonis et al., 2014)) as numerical values for our experiment instances. In this dataset, each unknown parameter is related to 8 features.

We use 210 instances for training and 90 instances for testing the model. For all proposed strategies and LP, we use a 5-layer fully connected network with 16 neurons per layer. We conduct experiments on 5 scales of the penalty factor ($\rho$): $\rho = 0.25, 0.5, 1, 2$, or $4$. Tables 7 and 8 report the mean post-hoc regrets and training times across 30 runs for each approach on WSMC respectively.

As Table 7 shows, the combination method consistently yields the smallest post-hoc regret, since it combines strengths from both strategies. MipOpt-Both achieves a post-hoc regret nearly identical to that of the combination method, with just around 1% higher regret across all penalty factors. The post-hoc regrets obtained by LCGT-Stage1 are noticeably larger than those of the combination method and MipOpt-Both, but they are still significantly better than vanilla LP.

Table 8 shows that LP consistently has the shortest training times as expected, while MipOpt-Both is somewhat slower. However, LCGT-Stage1 and "combination" are around 8-10x slower than LP.

To test whether the same effects/comparisons with training algorithms happen for other network architectures, we conducted an additional experiment with deeper networks (7 and 9 layers), using 10 simulations per penalty factor. The results show that the relative predictive-strength pattern persists for 7- and 9-layer networks, and that training time increases with depth for all methods, consistent with the added computational burden. See Appendix C for the detailed results and analysis.

In conclusion, while the combination method provides smallest post-hoc regret, it comes at a significantly higher computational cost. Given that MipOpt-Both achieves a similar level of post-hoc regret with much shorter training times, it is a strong alternative.

**0-1 Knapsack Problem** Our second benchmark is a variant of the 0-1 knapsack problem with unknown item prices $p_i$ and sizes $s_i$, which is a packing integer program. Since Hu et al. (2023) also experimented on this problem, we adopt their dataset and experimental setting. The problem description and formulation can be found in their Appendix C.2 (Hu et al., 2023). We use the same real data from Paulus et al. (2021) as that used in Hu et al. (2023) for the numerical values in our experimental instances. In this dataset, each 0-1 knapsack instance consists of 10 items and each item has 4096 features related to its price and size. For both our proposed strategies and LP, we use

Table 9: Full comparisons: mean **post-hoc-regret** and standard deviations (not confidence intervals) over 30 simulations on the 0-1 knapsack problem.

| Capacity | Penalty factor | LP | MipOpt-Both | LCGT-Stage1 | Combination |
|---|---|---|---|---|---|
| 100 | 0.21 | 1.70±0.49 | 1.26±0.01 | 1.28±0.13 | 1.26±0.01 |
| | 0.25 | 6.36±0.14 | 6.25±0.09 | 6.30±0.22 | 6.24±0.15 |
| | 0.3 | 10.06±1.55 | 9.53±0.57 | 9.16±0.67 | 9.11±0.21 |
| | 0.4 | 10.91±1.70 | 10.75±0.25 | 10.50±0.28 | 10.36±0.38 |
| 200 | 0.21 | 0.33±0.01 | 0.33±0.01 | 0.33±0.01 | 0.33±0.01 |
| | 0.25 | 1.67±0.03 | 1.67±0.03 | 1.67±0.03 | 1.67±0.03 |
| | 0.3 | 3.90±1.21 | 3.83±0.54 | 3.73±0.05 | 3.33±0.05 |
| | 0.4 | 6.99±0.26 | 6.79±0.26 | 6.70±0.22 | 6.66±0.10 |

a 5-layer fully-connected network with 512 neurons per hidden layer. We conduct experiments with 2 different knapsack capacities: 100 and 200. Following Hu et al. (2023), we also use 4 scales of the penalty factor: $0.21, 0.25, 0.3$, or $0.4$.

Table 10: Full comparisons: mean **training time** (in seconds) and standard deviations over 30 simulations on the 0-1 knapsack problem.

| Capacity | Penalty factor | LP | MipOpt-Both | LCGT-Stage1 | Combination |
|---|---|---|---|---|---|
| 100 | 0.21 | 219.78±28.65 | 390.83±75.41 | 791.05±144.58 | 745.57±156.54 |
| | 0.25 | 236.93±26.20 | 419.57±106.54 | 845.21±180.42 | 819.11±139.83 |
| | 0.3 | 286.13±94.82 | 421.58±152.69 | 844.05±158.99 | 826.00±173.35 |
| | 0.4 | 303.38±92.59 | 449.10±130.04 | 894.86±144.58 | 867.91±139.19 |
| 200 | 0.21 | 258.61±24.09 | 411.09±110.63 | 792.87±156.31 | 711.99±123.35 |
| | 0.25 | 290.36±25.79 | 433.61±97.58 | 804.04±175.93 | 794.52±136.67 |
| | 0.3 | 305.13±78.09 | 459.52±106.58 | 825.58±143.07 | 806.27±141.56 |
| | 0.4 | 328.74±106.52 | 466.97±125.98 | 859.78±195.93 | 812.03±100.89 |

We use 700 instances for training and 300 instances for testing the model performance. Table 9 compares the mean post-hoc regrets across 30 runs for each approach on the 0-1 knapsack problem. Given 0-1 knapsack is a much simpler problem than the other two problems we consider, the differences in post-hoc regret are not as large as in the other problem. As we increase the capacity of the knapsack, the integrality becomes less and less important in solutions, meaning there is less improvement possible over LP. We again see that while the best post-hoc regret comes from Combination, MipOpt-Both is never more than 5% worse than the combination. Table 10 compares the training times, LP as expected has the shortest training time, while MipOpt-Both requires around 50% longer. The combination requires close to twice the training time of MipOpt-Both for very little reduction in regret.

Again, the results point to MipOpt-Both as a strong practical alternative for handling integrality.

**Nurse Rostering Problem**  Our last benchmark is the nurse rostering problem (NRP). Consider a large medical center that needs to assign full-time nurses to shifts to meet patient load. If there are too many patients for full-time nurses assigned to a shift, the center can hire temp nurses at a higher salary to cover the extra demand. The task is to minimize the total hiring costs while meeting the patient demand and (full-time) nurse workload restrictions. The challenge is that the shifts need to be decided before the patient demand is known precisely, requiring prediction at the time of scheduling.

The roster of a particular week is made at least a full week before the start of the schedule, in order to let nurses plan their week. This is Stage 1, making a schedule using estimated patient demands.

The center requires patients to make appointments in advance—reservations for a week, from Monday to Sunday, close the Sunday night prior. That night, the center knows the precise patient load, and can solve the Stage 2 optimization to (A) potentially hire extra temp nurses to cover understaffed shifts or (B) edit full-time nurse schedules with a monetary compensation for their inconvenience.

Due to page limitations, see Appendix A.2 for a detailed prose description of the setup. The full Stages 1 and 2 MIP models and the Stage 2 penalty are also given in Appendix A.2.

We conduct experiments on the NSP with 10 full-time nurses. Extra nurses come at a cost of $\{15, 20, 25\}$ in different experiments. We again use real data from the ICON scheduling competition (Simonis et al., 2014) as the numerical values for patient demands. We use 70 instances for

Table 11: Full comparisons: mean **post-hoc-regret** and standard deviations (not confidence intervals) over 30 simulations on the nurse rostering problem.

| Extra payment | LP | MipOpt-Both | LCGT-Stage1 | Combination |
|---|---|---|---|---|
| 15 | 156.17±56.01 | 144.99±52.33 | 152.25±50.08 | 141.37±43.71 |
| 20 | 172.92±58.09 | 157.31±56.67 | 165.11±56.31 | 155.55±48.54 |
| 25 | 180.85±64.39 | 162.78±58.12 | 170.75±60.46 | 160.89±51.48 |

Table 12: Full comparisons: mean **training time** (in seconds) and standard deviations (not confidence intervals) over 30 simulations on the nurse rostering problem.

| Extra payment | LP | MipOpt-Both | LCGT-Stage1 | Combination |
|---|---|---|---|---|
| 15 | 1097.48±459.06 | 934.40±440.06 | 2031.60±782.70 | 1477.59±796.23 |
| 20 | 1299.55±555.45 | 1070.56±532.01 | 2353.85±790.74 | 1635.35±746.82 |
| 25 | 1570.99±696.68 | 1376.45±599.99 | 2695.65±865.36 | 1907.97±779.81 |

training and 30 instances for testing. We use a smaller dataset size for this benchmark since it is more complex and time-consuming to train models on than the previous two benchmarks. For all proposed strategies and LP, we use a 5-layer fully connected network with 16 neurons per hidden layer. Tables 11 and 12 report the mean post-hoc regrets and mean training times across 30 runs for each approach respectively.

From Table 11, we can see that the combination method again consistently outperforms all other methods, very closely followed by MipOpt-Both, while LCGT-Stage1 has prediction accuracy in between the combination method/MipOpt-Both and LP.

One interesting observation from Table 12 is that MipOpt-Both has even shorter training time than LP in this benchmark. This is because, although the training time for each epoch of MipOpt-Both is longer than that of LP, MipOpt-Both needs fewer epochs to converge. A similar trend holds between LimCut-Stage1 and the combination method as well.

In summary, MipOpt-Both again offers the best balance with essentially the best post-hoc regret, while having shorter training times, compared to other methods.

## 5 SUMMARY

The MIP-LP gap in Predict+Optimize has been an unresolved challenge in the community. In this paper, we propose a novel and *simple* strategy (MIPOpt, Section 3.2) to mitigate this gap, for the recently proposed Two-Stage Predict+Optimize framework. Compared with the slow adaptation of MIPaaL (Section 3.1), our experiments show that MIPOpt give the best tradeoff between predictive accuracy (in terms of post-hoc regret) and training time — it has essentially the best post-hoc regret, but has relatively mild runtime overhead compared to the vanilla LP-based gradient computation method. We thus advocate for the use of the simple strategy as an effective approach to incorporate integrality constraints into the prediction model training process.

We emphasize that the simplicity of our proposed novel strategy is a *key advantage*. Beyond the speed of the simple approach, it is also much easier to implement, and potentially applicable in a wider variety of settings (although this is work beyond the scope of this paper, requiring extensive further experimentation). Given that such a simple method is not an a-priori obvious strategy to try, we believe that our work is a practical and actionable result to disseminate to the community.

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

# A DESCRIPTIONS/MIP MODELS OF EXPERIMENTAL SETUPS

## A.1 WEIGHT SET MULTI-COVER

The Stage 1 solution is computed by solving the WSMC using the estimated coverage requirements $\hat{d}_i$:

$$\hat{x}^{(1)} = \arg\min_x \sum_{j \in J} c_j x_j$$
$$\text{s.t. } \sum_{j \in J} a_{ij} x_j \geq \hat{d}_i, \quad \forall i \in I$$
$$x_j \in \mathbb{N}, \quad \forall j \in J.$$

In Stage 2, the coverage requirements are revealed. To meet the coverage requirements, extra covers can be added with extra cost, for example, $\rho c_j$ for adding one cover $j$, where $\rho \geq 0$ is a non-negative tunable scalar parameter. In this scenario, the penalty function is:

$$Pen(\hat{x}^{(1)} \to x) = \rho c^\top (x - \hat{x}^{(1)})$$

With respect to the above penalty function, the Stage 2 solution is computed as:

$$\hat{x}^{(2)} = \arg\min_x \sum_{j \in J} c_j x_j + \rho \sum_{j \in J} c_j (x_j - \hat{x}_j^{(1)})$$
$$\text{s.t. } \sum_{j \in J} a_{ij} x_j \geq d_i \quad \forall i \in I$$
$$x_j \geq \hat{x}_j^{(1)}, \quad \forall j \in J$$
$$x_j \in \mathbb{N}, \quad \forall j \in J.$$

## A.2 NURSE ROSTERING PROBLEM

**Detailed description of the problem** Suppose there are $n$ full-time nurses, and we are scheduling a 7 day work week with 3 possible shifts per day, with the possibility of a day-off shift. Full-time nurses are entitled to take a rest: day-off shift. The unknown parameters are the numbers of patients that will come in each shift on each day next week $d \in \mathbb{R}^{7 \times 3}$. The decision variables are: 1) a Boolean vector $x \in \{0, 1\}^{n \times 7 \times 4}$, where $x_{i,j,k}$ represents that whether nurse $i$ is assigned to shift $k$ ($k \in \{1, 2, 3, 4\}$ with shift 4 denoting a day-off) in day $j$, and 2) an integer vector $\sigma \in \mathbb{N}^{7 \times 3}$, where $\sigma_{j,k}$ represents the number of extra nurses hired in shift $k$ day $j$. Let $d_{j,k}$ denote the number of patients in shift $k$ day $j$, $m_i$ denote the number of patients that the nurse $i$ can serve per shift, $c_i$ denote the payment of the nurse $i$ per shift, $e_s$ denote the number of patients that each extra nurse can serve per shift, and $e_c$ denote the payment of each extra nurse per shift. The unknown parameters are $d \in \mathbb{N}^{7 \times 3}$. The constraints are as follows: 1) the patient demand under each shift must be satisfied, 2) each full-time nurse is assigned to exactly one (working or rest) shift per day, 3) no full-time nurse may be scheduled to work a night shift followed immediately by a morning shift, and 4) each full-time nurse gets one or two day-off shifts in the week.

**Precise MIP models**   The Stage 1 solution is computed by solving the NRP using the estimations:

$$\hat{x}^{(1)}, \hat{\sigma}^{(1)} =$$

$$\arg\min_{x,\sigma} \sum_{i=1}^{n} c_i \sum_{j=1}^{7} \sum_{k=1}^{4} x_{i,j,k} + e_c \sum_{j=1}^{7} \sum_{k=1}^{3} \sigma_{j,k}$$

$$\text{s.t.} \sum_{i=1}^{n} m_i x_{i,j,k} + e_s \sigma_{j,k} \geq \hat{d}_{j,k}, \qquad \begin{aligned} &\forall j \in \{1,\ldots,7\}, \\ &k \in \{1,2,3\} \end{aligned}$$

$$\sum_{k=1}^{4} x_{i,j,k} = 1, \qquad \begin{aligned} &\forall i \in \{1,\ldots,n\}, \\ &j \in \{1,\ldots,7\} \end{aligned}$$

$$x_{i,j,3} + x_{i,j+1,1} \leq 1, \qquad \begin{aligned} &\forall i \in \{1,\ldots,n\}, \\ &j \in \{1,\ldots,6\} \end{aligned}$$

$$1 \leq \sum_{j=1}^{7} x_{i,j,4} \leq 2, \qquad \forall i \in \{1,\ldots,n\}$$

$$x \in \{0,1\}, \quad \sigma \in \mathbb{Z}.$$

In Stage 2, the center knows the precise number of patients for each shift in the upcoming week and can adjust the shift schedule, although this incurs additional costs. The additional costs are formulated as a penalty function defined above:

$$Extra(\hat{x}_{i,j,k}^{(1)} \to x_{i,j,k}) = \mathbf{1}\left[x_{i,j,k} > \hat{x}_{i,j,k}^{(1)}\right](T - j + 1)\rho_i c_i$$

Then we are ready to define the Stage 2 MIP:

$$\hat{x}^{(2)}, \hat{\sigma}^{(2)} =$$

$$\arg\min_{x,\sigma} \sum_{i=1}^{n} c_i \sum_{j=1}^{7} \sum_{k=1}^{3} x_{i,j,k} + e_c \sum_{j=1}^{7} \sum_{k=1}^{3} \sigma_{j,k}$$

$$+ \sum_{i=1}^{n} \rho_i c_i \sum_{j=1}^{7} \sum_{k=1}^{3} (T - j + 1)\gamma_{i,j,k}$$

$$\text{s.t.} \sum_{i=1}^{n} m_i x_{i,j,k} + e_s \sigma_{j,k} \geq d_{j,k}, \qquad \begin{aligned} &\forall j \in \{1,\ldots,7\}, \\ &k \in \{1,2,3\} \end{aligned}$$

$$\sum_{k=1}^{4} x_{i,j,k} = 1, \qquad \begin{aligned} &\forall i \in \{1,\ldots,n\}, \\ &j \in \{1,\ldots,7\} \end{aligned}$$

$$x_{i,j,3} + x_{i,j+1,1} \leq 1, \qquad \begin{aligned} &\forall i \in \{1,\ldots,n\}, \\ &j \in \{1,\ldots,6\} \end{aligned}$$

$$1 \leq \sum_{j=1}^{7} x_{i,j,4} \leq 2, \qquad \forall i \in \{1,\ldots,n\}$$

$$\gamma_{i,j,k} \geq x_{i,j,k} - \hat{x}_{i,j,k}^{(1)}, \qquad \begin{aligned} &\forall i \in \{1,\ldots,n\}, \\ &j \in \{1,\ldots,7\}, \\ &k \in \{1,2,3\} \end{aligned}$$

$$x \in \{0,1\}, \quad \sigma \in \mathbb{Z}, \quad \gamma \in \{0,1\}$$

## B   FULL EXPERIMENTS

In this appendix, we report the full experimental comparisons, including against traditional regression methods that were not designed for Predict+Optimize settings, including ridge regression (Ridge), $k$-nearest neighbors ($k$-NN), classification and regression tree (CART), random forest (RF), and neural network (NN).

**Weighted Set Multi-Cover Problem**   Table 13 is on the weighted set multi-cover problem, and presents the mean post-hoc regret and standard deviations (**not** confidence intervals, which are very difficult/impossible to do *correctly*) for various methods, categorized into three groups: the studied methods (our novel MipOpt-Both, the adaptation LCGT-Stage1 of MIPaaL, and Combination), the state-of-the-art Predict+Optimize method (LP), and weak non-Predict+Optimize methods. As explained in Section 4, "LP" is our only decision-focused baseline presented in this work. Most other decision-focused prior works cannot handle unknowns in constraints, and only CombOptNet (Paulus et al., 2021) and Nandwani et al. (2022) can in principle do it. However, Hu et al. (2023) already compared "LP" with CombOptNet and showed that "LP" is much better than the latter in prediction accuracy (in post-hoc regret), so we omit to repeat this comparison given how time consuming it is to run CombOptNet. As for Nandwani et al. (2022), even after corresponding with these authors we were unable to get a runnable version of their code, and so we do not compare with them.

Our studied methods (the adaptation of MIPaaL, our proposed novel MipOpt method, and the combination method) generally perform well, often resulting in the smallest 3 post-hoc regrets among all methods across different penalty factors. LP consistently yields larger post-hoc regrets than the proposed methods across all penalty factors but exhibits smaller post-hoc regrets compared to non-Predict+Optimize methods. Among non-Predict+Optimize methods, there is a notable variability in performance, for example, Ridge and RF perform better than $k$-NN and CART.

Table 13: Mean **post-hoc-regret** and standard deviations of Predict+Optimize and non-Predict+Optimize methods over 30 simulations on the weighted set multi-cover problem.

| Penalty factor | 0.25 | 0.5 | 1 | 2 | 4 |
|---|---|---|---|---|---|
| LP | 36.19±17.96 | 52.91±30.57 | 82.72±44.65 | 123.67±67.16 | 204.44±108.45 |
| MipOpt-Both | 30.70±16.12 | 49.78±27.48 | 76.16±46.55 | 118.18±62.40 | 188.53±92.73 |
| LCGT-Stage1 | 31.65±17.63 | 51.26±26.80 | 79.81±44.20 | 120.03±60.40 | 194.85±91.59 |
| Combination | 29.22±15.99 | 49.42±27.47 | 74.00±46.69 | 117.31±55.09 | 177.74±97.36 |
| Ridge | 42.40±24.95 | 53.69±31.22 | 84.27±44.20 | 124.07±72.34 | 211.72±124.50 |
| RF | 48.10±27.25 | 58.99±33.30 | 86.76±45.90 | 125.83±72.60 | 211.41±124.56 |
| NN | 47.61±27.66 | 58.62±33.41 | 86.64±45.47 | 124.68±70.46 | 212.77±121.38 |
| $k$-NN | 67.28±35.01 | 76.64±40.63 | 95.35±52.28 | 133.31±76.36 | 207.64±125.38 |
| CART | 63.26±32.81 | 74.64±39.19 | 97.42±52.81 | 142.96±81.42 | 234.05±140.18 |
| TOV | 346.33±200.37 | | | | |

**0-1 Knapsack Problem**   Table 14 reports the mean post-hoc regret and standard deviations for the studied methods, the state-of-the-art Predict+Optimize method (LP), and weak non-Predict+Optimize methods. As in the first benchmark, non–Predict+Optimize methods consistently perform worse than Predict+Optimize methods across different capacities and all penalty factors. In this benchmark, non–Predict+Optimize methods exhibit similar performance.

Table 14: Mean **post-hoc-regret** and standard deviations of Predict+Optimize and non-Predict+Optimize methods over 30 simulations on the 0-1 knapsack problem.

| Capacity | 100 | | | | 200 | | | |
|---|---|---|---|---|---|---|---|---|
| Penalty factor | 0.21 | 0.25 | 0.3 | 0.4 | 0.21 | 0.25 | 0.3 | 0.4 |
| LP | 1.70±0.49 | 6.36±0.14 | 10.06±1.55 | 10.91±1.70 | 0.33±0.01 | 1.67±0.03 | 3.90±1.21 | 6.99±0.26 |
| MipOpt-Both | 1.26±0.01 | 6.25±0.09 | 9.53±0.57 | 10.75±0.25 | 0.33±0.01 | 1.67±0.03 | 3.83±0.54 | 6.79±0.26 |
| LCGT-Stage1 | 1.28±0.13 | 6.30±0.22 | 9.16±0.67 | 10.50±0.28 | 0.33±0.01 | 1.67±0.03 | 3.73±0.05 | 6.70±0.22 |
| Combination | 1.26±0.01 | 6.24±0.15 | 9.11±0.21 | 10.36±0.38 | 0.33±0.01 | 1.67±0.03 | 3.33±0.05 | 6.66±0.10 |
| Ridge | 9.50±0.22 | 9.82±0.22 | 10.21±0.22 | 11.01±0.23 | 6.61±0.27 | 6.84±0.26 | 7.13±0.25 | 7.72±0.23 |
| RF | 9.53±0.26 | 9.85±0.26 | 10.25±0.27 | 11.05±0.29 | 6.63±0.31 | 6.86±0.29 | 7.15±0.28 | 7.73±0.25 |
| NN | 9.27±2.89 | 10.02±1.56 | 10.95±0.99 | 12.82±4.22 | 6.68±2.58 | 6.99±2.24 | 7.37±1.83 | 8.12±1.12 |
| $k$-NN | 9.46±0.22 | 9.77±0.22 | 10.27±0.22 | 11.96±0.23 | 6.45±0.22 | 6.69±0.22 | 6.99±0.21 | 7.60±0.21 |
| CART | 9.34±0.52 | 9.72±0.43 | 10.29±0.32 | 11.15±0.22 | 6.63±0.32 | 6.88±0.29 | 7.20±0.25 | 7.82±0.19 |
| TOV | 29.69±0.14 | | | | 48.14±0.17 | | | |

**Nurse Rostering Problem**   Table 15 is on the final nurse rostering problem benchmark. Consistent with the other two benchmarks, the table shows that non-Predict+Optimize methods consistently exhibit larger post-hoc regret compared to Predict+Optimize methods across all extra payment levels. Ridge and RF show the most competitive performance among the Non-Predict+Optimize methods, but still significantly lags behind LP and the proposed methods. This suggests that for complex optimization tasks like nurse rostering, incorporating the optimization problem into predictive models can significantly enhance performance.

Table 15: Mean **post-hoc-regret** and standard deviations of Predict+Optimize and non-Predict+Optimize methods over 30 simulations on the nurse rostering problem.

| Extra payment | 15 | 20 | 25 |
|---|---|---|---|
| LP | 156.17±56.01 | 172.92±58.09 | 180.85±64.39 |
| MipOpt-Both | 144.99±52.33 | 157.31±56.67 | 162.78±58.12 |
| LCGT-Stage1 | 152.25±50.08 | 165.11±56.31 | 170.75±60.46 |
| Combination | 141.37±43.71 | 155.55±48.54 | 160.89±51.48 |
| Ridge | 218.84±50.03 | 242.48±54.86 | 254.53±58.95 |
| RF | 219.96±48.77 | 241.71±53.12 | 252.66±53.44 |
| NN | 247.29±59.12 | 272.18±66.76 | 285.57±70.66 |
| $k$-NN | 235.24±43.86 | 256.34±51.44 | 270.70±54.13 |
| CART | 281.30±64.20 | 304.75±70.21 | 317.97±71.39 |
| TOV | 10598.58±1668.45 | 10812.51±1567.03 | 11021.66±1479.33 |

# C    ADDITIONAL EXPERIMENTS ON DEEPER NETWORK ARCHITECTURES

We ran a small experiment on the weighted set multi-cover problem, using 10 simulations per penalty factor, to assess whether the same relative strength pattern between training algorithms holds for deeper network architectures. We trained models with 7 and 9 layers and compared them with the 5-layer NN used in the main paper (we used this architecture because the SOTA (Hu et al., 2023) paper used it, and we wanted fair comparisons with them).

Table 16: Deeper network architectures: mean **post-hoc-regret** and standard deviations (not confidence intervals) over 10 simulations on the weighted set multi-cover problem.

| Penalty factor | | 0.25 | 0.5 | 1 | 2 | 4 |
|---|---|---|---|---|---|---|
| 5 layers | LP | 50.68±18.29 | 71.20±20.47 | 113.84±31.42 | 175.63±51.87 | 275.09±70.54 |
| | MipOpt-Both | 38.54±10.67 | 66.03±18.49 | 107.84±31.42 | 163.71±44.60 | 241.65±69.93 |
| | LCGT-Stage1 | 46.24±17.28 | 68.02±17.40 | 110.52±30.47 | 170.07±47.84 | 246.68±66.19 |
| | Combination | 38.29±10.61 | 65.29±18.24 | 106.33±30.88 | 161.87±45.94 | 240.79±68.09 |
| 7 layers | LP | 49.01±11.72 | 70.90±17.16 | 111.19±31.21 | 173.92±50.95 | 273.42±65.86 |
| | MipOpt-Both | 37.87±11.13 | 65.78±17.83 | 106.97±31.40 | 161.72±45.73 | 241.30±67.67 |
| | LCGT-Stage1 | 42.35±15.22 | 66.38±19.31 | 107.93±32.17 | 164.66±49.63 | 244.23±70.20 |
| | Combination | 37.75±10.72 | 64.89±18.16 | 106.11±29.94 | 160.79±45.70 | 240.28±69.08 |
| 9 layers | LP | 49.78±11.16 | 70.16±18.27 | 111.29±31.00 | 173.79±52.08 | 273.38±69.80 |
| | MipOpt-Both | 37.99±11.33 | 65.51±18.60 | 106.75±30.29 | 161.31±45.87 | 241.10±66.23 |
| | LCGT-Stage1 | 41.38±11.16 | 66.72±19.55 | 107.71±31.85 | 165.24±47.44 | 244.66±70.01 |
| | Combination | 37.65±11.92 | 64.83±18.24 | 106.13±31.95 | 160.85±45.62 | 240.62±69.27 |

As Table 16 shows, the pattern of relative predictive strengths between the training algorithms does continue to hold also for 7 layer and 9 layer networks: MipOpt-Both still outperforms the LP baseline and LCGT-Stage 1, and the Combination method has little to insignificant predictive accuracy gains over MipOpt-Both (at the expense of much longer training time).

In terms of the trends for the individual methods, increasing NN depth from 5 to 7 did improve the performance of all methods, including the LP baseline, with a slightly more pronounced gain for LCGT-Stage1. Increasing NN depth further from 7 to 9 layers did not substantially change the predictive accuracy for any training method.

Table 17: Deeper network architectures: mean **training time** (in seconds) and standard deviations (not confidence intervals) over 10 simulations on the weighted set multi-cover problem.

| Penalty factor | | 0.25 | 0.5 | 1 | 2 | 4 |
|---|---|---|---|---|---|---|
| 5 layers | LP | 91.93±22.98 | 89.05±33.21 | 100.15±54.98 | 78.87±27.02 | 61.74±14.17 |
| | MipOpt-Both | 143.94±52.41 | 159.05±63.21 | 183.32±69.28 | 170.31±65.54 | 137.22±43.45 |
| | LCGT-Stage1 | 788.26±300.46 | 803.54±334.65 | 880.17±353.09 | 773.56±321.68 | 723.19±346.35 |
| | Combination | 847.33±293.46 | 905.13±339.97 | 929.74±303.65 | 909.47±365.65 | 765.21±355.68 |
| 7 layers | LP | 96.26±23.34 | 92.00±24.73 | 113.71±58.80 | 85.97±36.73 | 84.74±24.32 |
| | MipOpt-Both | 159.84±54.21 | 192.60±66.58 | 203.71±88.99 | 197.87±72.53 | 169.00±51.92 |
| | LCGT-Stage1 | 800.96±294.42 | 858.53±363.96 | 895.84±377.92 | 807.15±307.92 | 773.64±396.95 |
| | Combination | 874.47±353.28 | 956.73±357.94 | 981.22±384.02 | 944.03±369.61 | 848.51±345.16 |
| 9 layers | LP | 98.93±21.62 | 96.54±38.20 | 117.95±59.07 | 90.77±27.86 | 89.64±25.65 |
| | MipOpt-Both | 166.20±57.25 | 210.80±69.48 | 218.73±83.91 | 199.85±67.67 | 172.32±61.25 |
| | LCGT-Stage1 | 835.05±305.34 | 885.97±365.92 | 905.45±367.48 | 836.21±326.91 | 799.25±401.61 |
| | Combination | 881.49±347.95 | 968.43±340.25 | 1007.49±395.13 | 971.95±390.24 | 872.04±335.46 |

Table 17 shows that training time increases with depth for all methods, consistent with the added computational burden.

In conclusion, these results show that (i) the relative advantages of MipOpt-Both and Combination over LCGT-Stage1 persist for deeper networks (and that Combination has minimal predictive accuracy advantage over MipOpt-Both but takes far longer training time), and (ii) accuracy gains from additional depth saturate beyond a moderate number of layers on WMSC. We will include these results if accepted.

## D  HYPERPARAMETERS FOR THE EXPERIMENTS

The methods of MipOpt-Both, LCGT-Stage1, Combination, LP, $k$-NN, RF, and NN have hyperparameters, which we tune via cross-validation: For the proposed methods and LP, we treat the optimizer, learning rate, the early-cut-off value of log barrier regularization term ($\mu$), and epochs as hyperparameters. For $k$-NN, we try $k \in \{1, 3, 5\}$; for RF, we try different numbers of trees in the forest $\{10, 50, 100\}$; for NN, we treat the optimizer, learning rate, and epochs as hyperparameters.

Tables 18, 19, and 20 show the final hyperparameter choices for the three problems: 1) weighted set multi-cover, 2) 0-1 knapsack problem, and 3) nurse rostering problem.

Table 18: Hyperparameters of the experiments on the weighted set multi-cover problem.

| Model | Hyperparameters |
|---|---|
| MipOpt-Both | optimizer: optim.Adam; learning rate: $10^{-6}$; $\mu = 10^{-3}$; warm start epochs = 20, stop epochs = 40 |
| LCGT-Stage1 | optimizer: optim.Adam; learning rate: $10^{-6}$; $\mu = 10^{-3}$; warm start epochs = 20, stop epochs = 40 |
| Combination | optimizer: optim.Adam; learning rate: $10^{-6}$; $\mu = 10^{-3}$; warm start epochs = 20, stop epochs = 40 |
| LP | optimizer: optim.Adam; learning rate: $5 \times 10^{-7}$; $\mu = 10^{-3}$; warm start epochs = 20, stop epochs = 40 |
| NN | optimizer: optim.Adam; learning rate: $10^{-4}$; stop epochs = 40 |
| $k$-NN | k = 5 |
| RF | number of estimator = 100 |

Table 19: Hyperparameters of the experiments on the 0-1 knapsack problem.

| Model | Hyperparameters |
|---|---|
| MipOpt-Both | optimizer: optim.Adam; learning rate: $10^{-6}$; $\mu = 10^{-3}$; warm start epochs = 5, stop epochs = 16 |
| LCGT-Stage1 | optimizer: optim.Adam; learning rate: $10^{-6}$; $\mu = 10^{-3}$; warm start epochs = 5, stop epochs = 16 |
| Combination | optimizer: optim.Adam; learning rate: $10^{-6}$; $\mu = 10^{-3}$; warm start epochs = 5, stop epochs = 16 |
| LP | optimizer: optim.Adam; learning rate: $10^{-7}$; $\mu = 10^{-3}$; warm start epochs = 5, stop epochs = 16 |
| NN | optimizer: optim.Adam; learning rate: $10^{-3}$; epochs=16 |
| $k$-NN | k = 5 |
| RF | number of estimator = 100 |

Ridge, $k$-NN, CART and RF are implemented using *scikit-learn* (Pedregosa et al., 2011). The neural network is implemented using *PyTorch* (Paszke et al., 2019).

## E  FURTHER RELATED WORK

Predict+Optimize is a relatively new and underexplored area, and therefore, there are very few works on mitigating the MIP-LP gap in this context, specifically addressing unknowns in constraints. Predict+Optimize falls under the broad umbrella of *decision-focused learning*, which includes works that learn prediction models for unknown parameters but with different goals/losses (Nandwani

Table 20: Hyperparameters of the experiments on the nurse rostering problem.

| Model | Hyperparameters |
|---|---|
| MipOpt-Both | optimizer: optim.Adam; learning rate: $10^{-5}$; $\mu = 10^{-7}$; warm start epochs = 12, stop epochs = 20 |
| LCGT-Stage1 | optimizer: optim.Adam; learning rate: ; $\mu =$; epochs= |
|  | optimizer: optim.Adam; learning rate: $10^{-5}$; $\mu = 10^{-7}$; warm start epochs = 12, stop epochs = 20 |
| LP | optimizer: optim.Adam; learning rate: $10^{-5}$; $\mu = 10^{-7}$; warm start epochs = 12, stop epochs = 20 |
| NN | optimizer: optim.Adam; learning rate: $10^{-4}$; stop epochs = 20 |
| $k$-NN | k = 5 |
| RF | number of estimator = 100 |

et al., 2022; Paulus et al., 2021). To better highlight our contributions, we provide an overview of how existing works —though not strictly within Predict+Optimize but still part of decision-focused learning—deal with the MIP-LP gap.

Some works (Wilder et al., 2019; Mandi & Guns, 2020; Hu et al., 2023) handle both LPs and ILPs by simply dropping the integrality constraints and use continuous-optimization based training methods. Such prior works do not use any integrality information in the underlying MIP, resulting in the MIP-LP gap that is addressed in the present work. Ferber et al. (2020) extends the work of Wilder et al. (2019) by using a cutting plane method to generate an LP problem that admits the same solution as the ILP as the relaxation problem. The relationship between our work and Ferber et al. (2020) is discussed in detail in the main paper, though the key issue is that Ferber et al. (2020) cannot handle uncertainty in constraints (and is much slower than the simple method proposed in this work).

On the other hand, some works (Pogančić et al., 2019; Sahoo et al.; Niepert et al., 2021; Berthet et al., 2020) consider optimization problems with a linear objective function, regardless of whether the problem contains discrete decision variables or not (i.e. they consider all of LPs, ILPs, and MILPs and beyond). Rather than viewing the optimization problem as a mapping from unknown parameters to a solution, these works interpret the optimization problem as mapping unknown parameters to probability distributions over the feasible set and optimizing the expected objective under this distribution. This probabilistic viewpoint yields differentiable surrogates that are agnostic to whether the variables are continuous or discrete, allowing for training with linear objectives without explicitly resolving the MIP–LP gap. However, none of these works can handle uncertainty in constraints, and as far as we can tell there does not seem to be any straightfoward adaptation to make these methods work with the Two-Stage Predict+Optimize framework.

