# OpenReview forum: "Efficient MIP-LP Gap Mitigation for Predict+Optimize"
_ICLR.cc/2026/Conference — Submitted to ICLR 2026_

### Official Review · Reviewer_1ivb · 2025-10-28

**Soundness:** 2
**Presentation:** 3
**Contribution:** 1
**Rating:** 4
**Confidence:** 3

**Summary:**

This paper examines how to address the MIP–LP gap in Predict+Optimize training, where integer constraints are typically relaxed. The authors propose a simple but effective fix: instead of using the LP relaxation solution when computing gradients, they directly use the true MIP optimum from neural solvers.

**Strengths:**

- The main idea is refreshingly simple. Instead of building a complex surrogate or adding a long chain of approximations, the authors just use the true MIP solution inside the training loop. The simplicity itself makes the message clear: sometimes the straightforward fix can go a long way.
- The paper reads clearly. The authors explain the setup step by step, which is not common in papers that integrate optimization and learning.
- The experiments are well organized and cover three different benchmarks. They show that this simple change can achieve accuracy on par with or slightly better than a more complex MIPaaL-style approach.

**Weaknesses:**

- The main concern is the limited novelty. The method is very close to MIPaaL and differs mostly in implementation. The paper does not introduce a new theoretical idea or learning principle.
- The use of the true MIP optimum is treated as a direct replacement, but the paper does not analyze why this replacement could contribute to more ideal gradients. The derivation in Section 3.2 depends on KKT conditions that are not valid for discrete problems, but this issue is not discussed.
- The use of KKT-based differentiation for a discrete problem is mathematically questionable. The KKT system assumes smoothness and convexity, which do not hold when integrality constraints are present. The paper admits this implicitly but does not discuss what kind of approximation the resulting gradient represents. This omission weakens the credibility of the method’s foundation.
- The method depends on solving a full MIP at every iteration. Although branch-and-bound solvers are faster than cutting-plane solvers, the runtime can still grow rapidly with problem size or solution variance. The paper does not analyze how training scales with instance complexity, nor does it discuss how solver randomness or multiple optimal solutions might affect gradient consistency.

**Questions:**

- When the discrete optimum changes between iterations, how is gradient discontinuity handled? Are there cases where training fails to converge?
- Since the gradient no longer comes from a differentiable mapping, what prevents the optimizer from following unstable directions?
- How does the proposed method handle multiple optimal MIP solutions? Which one is used for gradient computation, and does this choice affect the results?
- Can the authors clarify whether the gradients computed with integer solutions have any interpretation as subgradients of a convex envelope?
- How sensitive is the training process to solver tolerances and time limits? Could these parameters change the gradient signal significantly?

---

> ### Author Response · Authors · 2025-11-20
>
> Thank you for appreciating the (refreshing!) simplicity and effectiveness of our method. We address your comments/questions below.
>
> Weaknesses:
> 1. We first address a potential misunderstanding. Our final proposed method (the MIP optimum method in Section 3.2) crucially does not have anything to do with MIPaaL. Only the cutting plane method in Section 3.1, which we use to highlight the MIP optimum method we advocate (Section 3.2), was derived from MIPaaL.
>
> Our paper’s main conceptual message is that the simpler method has essentially no loss in prediction quality while being significantly faster than the MIPaaL adaptation. As the review recognized, simplicity is a virtue in this context and not a defect, since simpler methods are easier to apply and have the potential to be more widely applicable.
>
> 2, 3: We understand where the reviewer is coming from, but we strongly believe that our empirically-tested method that works well is worth disseminating to the ICLR community. While we do not have theoretical guarantees for our approach, many influential ML approaches that work empirically do not have any theoretical guarantees either (e.g. much of deep learning, beyond the ultra-simplified settings studied theoretically). In this paper, we opt to instead demonstrate that a simple technique works well in improving predictive accuracy. Given that our paper tackles an important problem in an important setting in Predict+Optimize, we believe that our method is still an empirically-successful and actionable technique that deserves dissemination to the community at ICLR.
>
> 4. Could the reviewer clarify the kind of analysis they are looking for? If it is worst-case analysis, the bounds are necessarily very pessimistic, and will depend on quantities that are close to impossible to bound, such as rate of convergence for optimizing non-convex problems. Moreover, as far as we know, both CPLEX and Gurobi use pseudo-random numbers to yield *deterministic* computations, so no true randomness is involved in the optimization.
>
> Questions:
>
> 1. Could the reviewer clarify what they mean by gradient discontinuity? In any gradient method, the “current point” changes, resulting in different gradients. As for the question about convergence, we did not observe any failure of convergence in our experiments.
>
> 2. Unfortunately we also do not follow this question. Could the reviewer further elaborate?
>
> 3. In our benchmarks, we simply use the optimum returned by Gurobi, for the MIP-optimum method.
>
> 4. There is no such interpretation, and that is kind of the point. Our work contrasts the SOTA Hu et al. paper: in their work, the gradients produced are indeed interpretable as some strongly-convexified version of the underlying MIP, but the strong convexification loses all integrality constraint information and hence really doesn’t have much to do with the underlying MIP anymore. On the other hand, in our proposed MIP optimum method, we reintroduce the integrality constraints via the MIP optimum itself, but the method loses any bona-fide gradient interpretation. We believe that the experimental results affirm our choice that despite the lack of a bona-fide gradient interpretation, our proposed learning method should be used given the superior predictive accuracy of the learned model.
>
> 5. For the MIP optimum method, we always allow Gurobi to solve to completion in our experiments. For the slow cutting plane method (which we use to highlight the MIP optimum method), Section 3.1 explores how solver tolerances change the learning behavior.

---

### Official Review · Reviewer_aJEK · 2025-10-30

**Soundness:** 3
**Presentation:** 3
**Contribution:** 3
**Rating:** 6
**Confidence:** 4

**Summary:**

This paper considers differentiable optimization for MIPs with both a first stage decision and then a recourse step (to repair feasibility violations at some cost), and where predictions enter into the constraints as well as the objective. They propose that instead of adding cutting planes to tighten the LP relaxation before differentiation, we can instead use a simpler straight-through style approach of substituting the exact MIP optimum into the LP before differentiating.

**Strengths:**

Predict + optimize with uncertainty is a challenging setting that most previous work avoids (preferring to handle only uncertainty in the objective). The proposed approach is very simple to implement and benefits from the (many) settings where MIPs are solvable efficiently via other strategies like branch and bound but not via cutting planes. It seems like a compelling drop-in replacement.

**Weaknesses:**

The experimental validation is not particularly thorough, particularly since 2 of the 3 settings use the same prediction task (which is from a separate domain unrelated to the optimization problem). The paper would be more convincing with more substantively different data distributions. The experiments are also all done with respect to a single, somewhat strange, architecture (5 layer neural network with 16 neurons per layer). Since predict+optimize methods often struggle with difficult training dynamics, it would be worthwhile to see if the findings generalize to other architectures.

**Questions:**

A broader set of experimental results are the biggest place for potential improvement.

---

> ### Author Response · Authors · 2025-11-20
>
> Thank you for your appreciation of our work, in particular its simplicity and usability.
>
> Regarding the **benchmark datasets**: this unfortunately remains an issue for this broader line of work, that there aren’t enough good datasets available. We agree with the reviewer and believe that the subfield should have a broader conversation on gathering datasets, but that is rather out of scope for the present work. Nonetheless, we strongly believe that our proposed method and results are still very actionable and worth disseminating to the ICLR community.
>
> Regarding the **neural network architecture**: we want to point out that, for 0-1 knapsack (see Appendix B), the architecture is actually 5 layers of **512** neurons each. We use this larger architecture due to the availability of more training data for that benchmark. Moreover, we also emphasize that we are just using the same architectures that Hu et al. already used in their “Two-Stage” paper for each benchmark, for a **consistent comparison**. Nonetheless, we agree with the reviewer that this is an interesting point, whether the same effects/comparisons with training algorithms happen for other network architectures. For the rebuttal period, we are aiming to complete an additional small experiment on the weighted multi-set cover problem, to demonstrate the effect of each training algorithm on (slightly) deeper neural networks. We hope to be able to upload something by the end of the author-reviewer discussion period.

---

> ### Author Response · Authors · 2025-12-01
>
> As promised, we ran a small experiment on the weighted set multi-cover problem, using only 10 simulations per penalty factor (due to time constraints in the discussion period), to assess whether the same relative strength pattern between training algorithms holds for deeper network architectures. We trained models with 7 and 9 layers and compared them with the 5-layer NN used in the main paper (we used this architecture because the SOTA Hu et al. paper used it, and we wanted fair comparisons with them).
>
> As Table 1 shows, the pattern of relative predictive strengths between the training algorithms does continue to hold also for 7 layer and 9 layer networks: MipOpt-Both still outperforms the LP baseline and LCGT-Stage 1, and the Combination method has little to insignificant predictive accuracy gains over MipOpt-Both (at the expense of much longer training time).
>
> In terms of the trends for the individual methods, increasing NN depth from 5 to 7 did improve the performance of all methods, including the LP baseline, with a slightly more pronounced gain for LCGT-Stage1. Increasing NN depth further from 7 to 9 layers did not substantially change the predictive accuracy for any training method.
>
> Table 1: mean post-hoc-regret and standard deviations (not confidence intervals) over 10 simulations on the WSMC problem.
> |  | Penalty   factor | 	0.25	| 	0.5 	|   	1  	|   	2  	|   	4  	|
> |:----------------:|:------------:|:-----------:|:-----------:|:------------:|:------------:|:------------:|
> |	5   layers	|  	LP  	| 50.68±18.29 | 71.20±20.47 | 113.84±31.42 | 175.63±51.87 | 275.09±70.54 |
> |      	| MipOpt-Both | 38.54±10.67 | 66.03±18.49 | 107.84±31.42 | 163.71±44.60 | 241.65±69.93 |
> |       |  LCGT-Stage1 | 46.24±17.28 | 68.02±17.40 | 110.52±30.47 | 170.07±47.84 | 246.68±66.19 |
> |       |  Combination | 38.29±10.61 | 65.29±18.24 | 106.33±30.88 | 161.87±45.94 | 240.79±68.09 |
> |	7   layers	|  	LP  	| 49.01±11.72 | 70.90±17.16 | 111.19±31.21 | 173.92±50.95 | 273.42±65.86 |
> |       | MipOpt-Both | 37.87±11.13 | 65.78±17.83 | 106.97±31.40 | 161.72±45.73 | 241.30±67.67 |
> |       |  LCGT-Stage1 | 42.35±15.22 | 66.38±19.31 | 107.93±32.17 | 164.66±49.63 | 244.23±70.20 |
> |       |  Combination | 37.75±10.72 | 64.89±18.16 | 106.11±29.94 | 160.79±45.70 | 240.28±69.08 |
> |	9   layers	|  	LP  	| 49.78±11.16 | 70.16±18.27 | 111.29±31.00 | 173.79±52.08 | 273.38±69.80 |
> |       | MipOpt-Both | 37.99±11.33 | 65.51±18.60 | 106.75±30.29 | 161.31±45.87 | 241.10±66.23 |
> |       |  LCGT-Stage1 | 41.38±11.16 | 66.72±19.55 | 107.71±31.85 | 165.24±47.44 | 244.66±70.01 |
> |       |  Combination | 37.65±11.92 | 64.83±18.24 | 106.13±31.95 | 160.85±45.62 | 240.62±69.27 |
>
> Table 2 shows that training time increases with depth for all methods, consistent with the added computational burden.
>
> Table 2: mean training time (in seconds) and standard deviations (not confidence intervals) over 10 simulations on the WSMC problem.
> |  |Penalty   factor|  	0.25 	|  	0.5  	|    	1	   |   	2   	|   	4   	|
> |:----------------:|:------------:|:-------------:|:-------------:|:--------------:|:-------------:|:-------------:|
> |	5   layers	|  	LP  	|  91.93±22.98  |  89.05±33.21  |  100.15±54.98  |  78.87±27.02  |  61.74±14.17  |
> |      | MipOpt-Both |  143.94±52.41 |  159.05±63.21 |  183.32±69.28  |  170.31±65.54 |  137.22±43.45 |
> |      |  LCGT-Stage1 | 788.26±300.46 | 803.54±334.65 |  880.17±353.09 | 773.56±321.68 | 723.19±346.35 |
> |      |  Combination | 847.33±293.46 | 905.13±339.97 |  929.74±303.65 | 909.47±365.65 | 765.21±355.68 |
> |	7   layers	|  	LP  	|  96.26±23.34  |  92.00±24.73  |  113.71±58.80  |  85.97±36.73  |  84.74±24.32  |
> |      | MipOpt-Both |  159.84±54.21 |  192.60±66.58 |  203.71±88.99  |  197.87±72.53 |  169.00±51.92 |
> |      |  LCGT-Stage1 | 800.96±294.42 | 858.53±363.96 |  895.84±377.92 | 807.15±307.92 | 773.64±396.95 |
> |      |  Combination | 874.47±353.28 | 956.73±357.94 |  981.22±384.02 | 944.03±369.61 | 848.51±345.16 |
> |	9   layers	|  	LP  	|  98.93±21.62  |  96.54±38.20  |  117.95±59.07  |  90.77±27.86  |  89.64±25.65  |
> |      | MipOpt-Both |  166.20±57.25 |  210.80±69.48 |  218.73±83.91  |  199.85±67.67 |  172.32±61.25 |
> |      |  LCGT-Stage1 | 835.05±305.34 | 885.97±365.92 |  905.45±367.48 | 836.21±326.91 | 799.25±401.61 |
> |      |  Combination | 881.49±347.95 | 968.43±340.25 | 1007.49±395.13 | 971.95±390.24 | 872.04±335.46 |
>
> In conclusion, these results show that (i) the relative advantages of MipOpt-Both and Combination over LCGT-Stage1 persist for deeper networks (and that Combination has minimal predictive accuracy advantage over MipOpt-Both but takes far longer training time), and (ii) accuracy gains from additional depth saturate beyond a moderate number of layers on WMSC. We will include these results if accepted.

---

### Official Review · Reviewer_9gep · 2025-10-31

**Soundness:** 2
**Presentation:** 2
**Contribution:** 2
**Rating:** 2
**Confidence:** 4

**Summary:**

This paper studies decision-focused learning for MILP and addresses the mismatch between training on continuous LP relaxations and testing on discrete MIP problems. Thus, The authors adopt a two-stage predict–then–optimize formulation, where an initial decision based on predicted parameters is later adjusted once the true parameters—appearing in both the objective and the constraints—are revealed, with a linear penalty capturing the cost of revision.

Two training strategies are proposed: adding cutting planes to tighten LP relaxations, and using the true MIP optimum instead of the LP solution during gradient computation.

Experiments on Weighted Set Multi-Cover, 0–1 Knapsack, and Nurse Rostering demonstrate that the latter approach achieves comparable decision quality to more complex combinations while being significantly faster, providing a practical balance between accuracy and efficiency.

**Strengths:**

- **Originality:** The proposed strategy, which leverages non-stationary KKT information from integer optima as surrogate gradients, is simple yet novel in combining exact discrete solutions with differentiable LP-based learning.
- **Quality:** The experimental evaluation covers three benchmark problems of different structure and scale. Results consistently show that using the true MIP optimum improves decision quality over standard LP-based training. The comparisons against cutting-plane variants are fair and well-motivated.
- **Clarity:**  The paper is clearly written and easy to follow, with a logical presentation of the two-stage framework and experimental methodology.
- **Significance:** The work addresses a practically important limitation in the gap between LP relaxation and MILP. The proposed approach offers a pragmatic balance between accuracy and efficiency and could influence future work on learning-integrated optimization for MIPs and related combinatorial problems.

**Weaknesses:**

- **Missing related work:** The paper frames the problem as “predict + optimize” but overlooks a large body of existing research under the umbrella of decision-focused learning (DFL). Several prior works have already considered mixed-integer problems, such as SPO+ [1] and PFYL [2]. While many of these methods focus on predicting objective coefficients rather than constraint parameters, their formulations and theoretical insights remain directly relevant to the proposed approach. The absence of this discussion creates the impression that the contribution is more novel than it actually is, and the paper would benefit from explicitly positioning itself within the established DFL literature.
- **Lack of theoretical justification:** The proposed “MIP Optimum Replacement” strategy injects integer solutions into the KKT system to compute surrogate gradients. However, this process is inherently non-stationary and discontinuous, since integer optima do not satisfy the first-order optimality conditions of the continuous relaxation. The paper provides no theoretical justification, convergence argument, or stability analysis to support the validity of such gradients. This concern becomes even more critical when the LP relaxation and MIP solution differ substantially, as the resulting gradients may no longer reflect meaningful descent directions and could introduce significant bias or instability during training.
- **Strong Assumption Under Ad-hoc penalty:** The paper addresses infeasible predictions through an ad-hoc linear adjustment penalty proportional to the cost coefficients. This relies on a strong assumption that incorrect decisions can be revised and that the cost of revision scales directly with the original objective weights. While computationally convenient, such a design lacks theoretical or economic justification and is not a standard practice in optimization or decision analysis.
- **Limited experimental scale and generality:** The evaluation is restricted to small, toy-sized benchmarks with only tens of variables and constraints. For instance, the Knapsack and Nurse Rostering problems involve fewer than 50 decision variables, which limits the relevance of the reported improvements to realistic settings. As a result, it remains unclear whether the proposed method would remain stable or computationally viable on larger or more structured mixed-integer problems.

[1] Mandi, J., Stuckey, P. J., & Guns, T. (2020, April). Smart predict-and-optimize for hard combinatorial optimization problems. In Proceedings of the AAAI conference on artificial intelligence (Vol. 34, No. 02, pp. 1603-1610).

[2] Berthet, Q., Blondel, M., Teboul, O., Cuturi, M., Vert, J. P., & Bach, F. (2020). Learning with differentiable pertubed optimizers. Advances in neural information processing systems, 33, 9508-9519.

**Questions:**

1. The proposed method uses integer optima within the KKT system to compute gradients, which are inherently non-stationary. Have the authors analyzed whether these surrogate gradients provide unbiased or stable updates in expectation?
2. Intuitively, the effectiveness of the MIP Optimum Replacement strategy should depend strongly on the tightness of the LP relaxation. Have the authors evaluated how the method behaves when the relaxation is loose and the integrality gap between the LP and MIP optima is large? In such cases, could the substituted KKT gradients become less informative or even detrimental to training performance?
3. The experiments are conducted on small-scale instances with fewer than 50 variables. Could the authors comment on the computational scalability of their approach for larger problems?
4. The paper employs CPLEX for cutting-plane experiments and Gurobi for MIP optimization. Have the authors verified that solver-specific behaviors do not bias the comparison in runtime or solution quality?

---

> ### Author Response · Authors · 2025-11-20
>
> Thank you for your review. We address your comments below.
>
> Weaknesses:
>
> 1. Both the papers referenced in the review are already cited in our paper. We also have Appendix D discussing "Further Related Work", including an expanded discussion relating to the broader Decision-Focused Learning literature.
>
> 2. Indeed, we do not have any theoretical guarantees for our approach. On the other hand, many ML approaches that work empirically do not have any theoretical guarantees either, including many influential algorithms (e.g. much of deep learning, beyond the very-simplified settings studied theoretically). More closely related to our work, the SOTA paper by Hu et al. constructs training gradients that are indeed the gradients for some strongly-convexified relaxation of the MIP, but the convexification crucially loses all the information about the integrality constraints. So, despite having theoretical guarantees, these guarantees are not actually directly relevant to the learning problem at hand. In this paper, we instead opt to demonstrate that a simple technique works well in improving predictive accuracy. Despite not having full theoretical justification, we believe that it is still an empirically-successful and actionable technique that deserves dissemination to the community at ICLR.
>
> 3. The reviewer’s criticism seems to be directed at the established Two-Stage Predict+Optimize framework introduced by Hu et al. Here, we point out again that the penalty function is designed specifically for each application, and is tailored towards the economics of that application. Moreover, such a recourse+penalty framework is reminiscent of the classic Two-Stage Stochastic Optimization framework, which is standard and (very) well-established in the stochastic optimization literature.
>
> In addition, the reviewer might have a misunderstanding regarding the specific penalty functions used in our benchmarks. It is not true that we only used restrictive linear penalties that scale only with the size of objective coefficients. For example, see our Nurse Rostering formulation: the penalty function there allows for a penalty factor that is different for each nurse. This is the most general form of additive penalty.
>
> 4. Our Nurse Rostering benchmark includes 301 decision variables, which is much larger than 50, in the context of MIPs.
>
> Questions:
>
> 1. The gradients in the context of the MIP is going to be 0 almost everywhere (and hence completely uninformative), and the line of work that this paper sits in is to generate useful training gradient signals even if they are not the actual gradients of the MIP solution with respect to its parameters. Along this line of reasoning, if one were to design surrogate gradients that are unbiased, then they will have expectation 0 (in some almost-everywhere sense) and again not useful at all. Could the reviewer clarify what they are asking?
>
> 2. The main purpose of our paper is to mitigate the MIP-LP gap. As such, our methods are designed to fix the performance deficiencies of the SOTA (Hu et al.), when the LP relaxations are loose compared to the original MIPs. Our experiments confirm that our proposed method (the fast, MIP optimum method) indeed outperforms the SOTA.
>
> 3. As mentioned above, our largest-scale benchmark (Nurse Rostering) includes 301 decision variables. These experiments were all run on the free version of Google Colab given that we have rather limited computational resources ourselves, and so we believe that there will be further scaling available using more advanced hardware.
>
> 4. In a way, the solving-algorithm specific behavior is one of the main points we make, when comparing the slow cutting plane method vs our fast MIP optimum method. Cutting plane solvers are inherently slower than SOTA MIP solvers (which tend to use branch-and-bound as the main component of the algorithm). As stated on Page 2, our fast MIP optimum method avoids needing cutting plane solvers, which is the main reason why it runs so much faster than the cutting plane method.

---

### Official Review · Reviewer_eSbT · 2025-11-01

**Soundness:** 3
**Presentation:** 3
**Contribution:** 2
**Rating:** 4
**Confidence:** 3

**Summary:**

This paper proposes new methodologies for solving MIP in the predict+optimize paradigm beyond directly solving the relaxed continuous version of the optimization problem.

Specifically, the authors first proposed a cutting-plane-based method that aims to solve the MIP directly but has high time complexity. To address this complexity issues, they also proposed a faster method that aims to solve the MIP objective directly. Additionally, the authors use numerical results to illustrate the performance and interpret some intuitions.

**Strengths:**

1. The paper is very clear in its setup, contributions, and results.
2. The paper proposed many new methods of solving MIP in the Predict+Optimize framework, instead of solving the relaxed problem directly.

**Weaknesses:**

1. The exact methodology of choosing cutting planes corresponding to Section 3.1, as well as the branch-and-bound methodology, seems not to be included in the paper. Could the authors elaborate more on that?

2. It seems the paper is purely experimental. Is there any theoretical guarantee on the performance? I am asking this question because both (1) the choice of cutting plane and (2) branch-and-bound choices may depend on the assumption of approximation in Stage 1, so any theoretical results will be greatly appreciated.

3. If this paper wants to purely focus on experimental results, only two problem instances might not be strong enough to answer the listed questions in the paper.

**Questions:**

1. Conceptually, what are the advantages of using the cutting-plane-based method?

2. If the paper advocates for the faster method, I wonder why the authors still propose the slow one and use it as a benchmark, given it is not an SOTA algorithm.

3. Please also see my questions in the weaknesses section.

---

> ### Author Response · Authors · 2025-11-20
>
> Reviewer eSbt:
>
> Thank you for your constructive questions. Below, we address the comments/weaknesses/questions in your review.
>
> Summary:
> We want to point out that our method is for **training** a neural network to predict unknown parameters in a MIP (including in its constraints). The focus is not on solving MIPs. Solving MIPs is part of the training process but is not the overall challenge.
>
> Weaknesses:
> 1. We configured CPLEX to be a pure cutting plane solver, but otherwise, we just use whatever CPLEX implemented by default (which generates a variety of cuts including Gomory Cuts, Gomory Fractional Cuts, Mixed Integer Rounding Cuts and Disjunctive Cuts). Similarly, we just run Gurobi using default settings. We believe it is important in our experiments that we make use of the defaults of the commercial solvers, reflecting the common usage patterns of these solvers by general practitioners.
>
> 2. This paper is indeed experimental and empirical in nature, and unfortunately it is extremely challenging to prove (useful and relevant) theory in this complex setting involving both non-convex MIPs in addition to learning dynamics. We do point out that the same issue holds for most if not all directly-relevant works inspiring our paper though: for example, the SOTA Hu et al. paper constructs training gradients that are indeed the gradients for some strongly-convexified relaxation of the MIP, but the convexification crucially loses all the information about the integrality constraints. In this paper, we instead opt to demonstrate that a simple technique works well in improving predictive accuracy. Despite not having full theoretical justification, we believe that it is still an actionable technique that deserves dissemination to the community at ICLR. This is particularly important as our paper tackles an important problem in an important setting in Predict+Optimize.
>
> 3. Our paper has 3 benchmarks, not just 2. Due to submission page limits, we put the 0-1 knapsack benchmark in Appendix B, and only gave a brief forward reference in the main text. We understand that it might have been easy to miss that forward reference, and so, if accepted, we will move key parts of the results for that benchmark to the main body using the allowed extra page.
>
> Questions:
>
> 1. Compared to the original SOTA Hu et al. paper, the use of the cutting plane method reduces the (conceptual) integrality gap by better approximating the (convex hull of the) MIP feasible set.
>
> 2. There are two answers, one conceptual and one meta/pragmatic. Conceptually, the cutting plane method *does* yield more accurate predictions than Hu et al.’s SOTA method, in terms of the post-hoc regret, although at the expense of much longer training time. So we decided to include it for that reason. Pragmatically, if we didn’t include the cutting plane method, a reviewer who is familiar with this literature would probably ask why we didn’t try adapting MIPaaL for the Two-Stage setting, so for this reason we also included the method.
>
> We also point out that, among the previous methods that deal with MIP-LP gaps in Predict+Optimize with uncertainty only in the **objective**, MIPaaL was the only method that seems straightforwardly adaptable to the Two-Stage setting with uncertainty in the constraints as well. This is another reason why we chose to present it in this paper.
>
> We definitely welcome suggestions on the writing, if you think there are things we should further clarify!

---

### Author Response · Authors · 2025-12-03
**Updated paper revision**

We have updated the paper as promised, now including (1) reorganizing the paper to make the 0-1 knapsack benchmarks easier to find, that the paper includes 3 different benchmarks, and (2) our followup experiments as suggested by Reviewer aJEK, experimenting on deeper NN architectures (beyond our direct comparison with the SOTA Hu et al. paper using their shallower architecture), where we show that same relative strengths comparing the different training algorithms.

---

### Meta-Review · Area_Chair_fajJ · 2026-01-07

**Summary:**

This paper addresses the precision loss problem caused when mixed-integer linear programming is relaxed into linear programming, and proposes a simple and fast method to solve the mixed-integer linear programming-linear programming gap. However, the reviewers have concerns about the theoretical guarantees of this method and its generalization to large-scale datasets, which the authors did not respond well to. Therefore, I recommend rejecting this article.

**Reviewer Concerns:**

All four reviewers did not respond. Among them, the 6-point reviewer suggested validating the effectiveness of the method on more types of datasets, but the authors refused this due to the lack of high-quality data. Of the two 4-point reviewers, one believed that the work had limited novelty and that some of the proposed theoretical issues were not resolved. The related proof issues of the other reviewer were rejected by the authors, while other issues were resolved. Additionally, The theoretical guarantee issue of the 2-point reviewer was not resolved, some large-scale extended supplementary experiments were also not provided by the authors, and the authors hope to have further discussion on some other issues.

**Reviewer Scores:**

I think none of the four reviewers will increase the score, as the paper still has some core issues unresolved, such as theoretical guarantees and generalizability. The final score is still 4,2,6,4

---

### Decision · Program_Chairs · 2026-01-26

Reject